# Morphophysiological and Proteomic Responses on Plants of Irradiation with Electromagnetic Waves

**DOI:** 10.3390/ijms222212239

**Published:** 2021-11-12

**Authors:** Zhuoheng Zhong, Xin Wang, Xiaojian Yin, Jingkui Tian, Setsuko Komatsu

**Affiliations:** 1College of Biomedical Engineering & Instrument Science, Zhejiang University, Hangzhou 310027, China; zhongzhh@zju.edu.cn (Z.Z.); tjk@zju.edu.cn (J.T.); 2College of Agronomy and Biotechnology, China Agricultural University, Beijing 100193, China; 020350286@163.com; 3Department of Pharmacognosy, China Pharmaceutical University, Nanjing 211198, China; ajian.517@163.com; 4Faculty of Environmental and Information Sciences, Fukui University of Technology, Fukui 910-8505, Japan

**Keywords:** proteomics, crop, millimeter waves, ultraviolet, gamma ray

## Abstract

Electromagnetic energy is the backbone of wireless communication systems, and its progressive use has resulted in impacts on a wide range of biological systems. The consequences of electromagnetic energy absorption on plants are insufficiently addressed. In the agricultural area, electromagnetic-wave irradiation has been used to develop crop varieties, manage insect pests, monitor fertilizer efficiency, and preserve agricultural produce. According to different frequencies and wavelengths, electromagnetic waves are typically divided into eight spectral bands, including audio waves, radio waves, microwaves, infrared, visible light, ultraviolet, X-rays, and gamma rays. In this review, among these electromagnetic waves, effects of millimeter waves, ultraviolet, and gamma rays on plants are outlined, and their response mechanisms in plants through proteomic approaches are summarized. Furthermore, remarkable advancements of irradiating plants with electromagnetic waves, especially ultraviolet, are addressed, which shed light on future research in the electromagnetic field.

## 1. Introduction

Electromagnetic waves are nonchemical waves and move with speed equal to the speed of light in a vacuum condition. According to different frequencies and wavelengths, electromagnetic waves are typically divided into eight spectral bands, including audio waves, radio waves, microwaves, infrared, visible light, ultraviolet (UV), X-rays, and gamma rays [1]. These electromagnetic waves are classified into either a non-ionizing irradiation or ionizing irradiation category [2]. The electromagnetic spectrum has potential applications in plant protection; however, interactions between electromagnetic waves and organisms largely rely on wave frequency and penetration depth of waves in the medium [1]. Herein, the electromagnetic spectrum, wavelength, frequency, as well as the source to emit electromagnetic-wave bands are summarized (Table 1).

In the agricultural area, electromagnetic-wave irradiation has been used to develop crop varieties, manage insect pests, monitor fertilizer efficiency, and preserve agricultural produce [4]. Effects of electromagnetic-wave irradiation on seed germination, seedling establishment, and crop productivity have been investigated in wheat [5], rice [6], maize [7], soybean [8], and sugar beet [9]. In rice, UV-B irradiation suppressed seedling growth during leaf development [6]; however, microwave irradiation of soils increased yields by 34% compared to the control [10]. Influences of electromagnetic-wave irradiation on crops are dose dependent, where low doses induced fewer side effects compared with high doses that affected plant phenotype. UV RESISTANCE LOCUS 8 (UVR8) photoreceptor, which played critical roles in phenotypic diversity under UV-B irradiation, mediated regulatory response of flavonoids/phytohormones or activated stress response of cell cycle under low or high dose UV-B, respectively [11]. These studies presented the two sides of electromagnetic-wave irradiation on crops and a serial of cellular metabolisms that were activated in a dose-dependent manner of electromagnetic-wave irradiation.

In response to long-term exposure to gamma irradiation, more fractions of protein, lipid, amino acid, and polysaccharides were detected in black gram seeds than control, which correlated with increasing biochemical metabolites [12]. Sucrose content sharply declined in rice leaves exposed to short-term UV-B [6]; however, soluble sugar continuously accumulated in wheat leaves irradiated with gamma rays during the first 4-week treatment [5]. Although, UV-B irradiation resulted in generation of free radicals, which induced conversion of amino acids such as gamma-aminobutyric acid and glutathione into antioxidants [13]. In plants, the contents of secondary metabolites fluctuated by electromagnetic-wave irradiation and accumulation of photosynthetic pigment, flavonoid, polyamine, and alkaloid were augmented by UV [14,15,16]. It was reviewed that phytohormones participated in UV-mediated responses via the UVR8-dependent or independent signaling pathway [17,18]. These findings proved that electromagnetic-wave irradiation affected metabolite profiles in plants, while receptors to a specific electromagnetic spectrum need mining to construct signaling pathways in profiling metabolite shift.

Millimeter waves, UV-B, and gamma rays improved plant tolerance against salt [19,20], drought [21,22], flooding [23,24], and biotic stresses [8,25] through activation of stress responsive pathways associated with redox signaling, carbohydrate metabolism, and ion homeostasis. UV-B strengthened tobacco drought tolerance via increasing flavonoid in leaves [22], and gamma rays aggravated sugarcane salt stress via accumulation of salt ions and osmolytes [19]. Soybean seeds pretreated with millimeter waves or gamma rays promoted seedling growth under flooding stress, and proteins involved in glycolysis, fermentation, and the cell wall played roles to counterbalance flood stimuli [23,24,26]. In addition, several studies showed that other sources of electromagnetic waves, such as microwaves, exerted opposite impacts on crop physiology dependent on plant varieties and exposure time, especially for seed germination, which was concluded by examining germination rate, shoot length, and plant biomass [27,28]. However, apart from these morphological changes, physiological changes and regulatory mechanisms provoked by audio waves and microwaves were far behind what have been obtained from millimeter waves, ultraviolet, and gamma rays. Meanwhile, proteomics utilized alone or integrated with other omics approaches could provide a deeper knowledge of different plant processes [29]. Thus, in this review, effects of millimeter waves, UV, and gamma rays on plants are outlined and electromagnetic-wave-mediated plant responses through proteomic approaches are summarized. Furthermore, remarkable advancements of irradiating plants with electromagnetic waves, especially ultraviolet, are addressed, which shed light on future research in the electromagnetic field.

## 2. Millimeter Waves

### 2.1. Characteristics

In the electromagnetic spectrum, millimeter waves position at the overlap between microwaves and infrared. The radio frequency of millimeter waves extends from 30 to 300 GHz, corresponding to wavelengths from 10 to 1 mm. The photon energy of millimeter waves could not ionize molecules, which made millimeter-wave irradiation an environmentally appropriate technique and led to small adverse effects to human health [30]. Millimeter waves are virtually absent from the natural electromagnetic environment [31]. Compared to wave bands with lower frequencies, millimeter waves have a shorter wavelength, which allows modest size antennas to have a smaller beam width and a greater frequency reuse potential. However, the higher frequency of millimeter waves leads to higher atmosphere attenuation, precipitation attenuation, diffraction effects, and scattered effects, which means millimeter waves are largely absorbed by gases or humidity in the environment and are bad at penetrating solid material and traveling long distance [32]. Nevertheless, compared to optical waves that have higher frequencies, atmosphere attenuation of millimeter waves is much less, and more importantly, the propagation of millimeter waves is less influenced by the lights and thermal effects from the environment [33].

These characteristics have seen the potential of millimeter waves being applied to telecommunications [34], weapons-systems making [35], security screening [36], and medical diagnosis [37]. One of the contemporary applications of millimeter waves is for construction of the newest generation of cell phone networks, which are known as 5G networks, using frequencies which range near the bottom of the bands [38]. Since millimeter waves are absent from the natural electromagnetic environment, living organisms might lack the adaption to them during evolution [31]. The growing use of millimeter waves in practical applications make it imperative to gain comprehensive knowledge about their bioeffects for health hazard evaluation, giving rise to the interest in the biological research that relates to millimeter waves.

### 2.2. Morphophysiological Effects

Owing to the characteristics, millimeter-wave irradiation is an environmentally appropriate technology with small threats to human health, which is important for sustainable development and worthy for research on their effects. First of all is the induction of thermal energy into the biological system via incident irradiation, which resulted in local heating of water molecules in surface cell membranes [39]. Moreover, many non-thermal effects of millimeter-wave irradiation were discovered. It was revealed that optimum millimeter-wave irradiation stimulated cell division, enzyme synthesis, growth rate, and biomass yield of a variety of microorganisms [40]. Such morphophysiological effects of millimeter-wave irradiation on microorganisms encouraged the investigation of applying irradiation on crops for increasing crop productivity in the agricultural industry [41].

Non-thermal effects of millimeter-wave irradiation targeting crop seeds have been studied and stimulatory effects on crop morphology were summarized in Table 2, including wheat, chickpea, soybean, and brown rice. The effects of millimeter-wave irradiation on wheat were mostly studied. Millimeter-wave irradiation on wheat seed at the initial stage not only improved germination [42,43], but also for the subsequent growth of shoot and grains [44,45]. In crop brown rice, similar effects of millimeter-wave irradiation on germination were found and content of polyphenols increased, while gamma-aminobutyric acid decreased [46]. Over the recent two years, the effects of millimeter-wave irradiation on leguminous plants have been studied. Many plants in the Leguminosae family, such as soybean and chickpea, are an important source for vegetable proteins; however, their growth was easily affected by flooding stress [47]. Irradiating soybean seeds and chickpea seeds with millimeter waves improved plant growth and tolerance under flooding as well, which might be a feasible approach for development of stress-tolerant lines and have benefits for crop yield [24,48].

Apart from plant seeds, other plant organs also responded to millimeter-wave irradiation. In the pollen grains of kiwifruit treated with indirect millimeter-wave irradiation, which used water that was irradiated with 40–78 GHz of millimeter waves for preparing the growth medium, pollen tube growth increased immediately and continuously increased until several days after the treatment compared with untreated groups [49]. In another study which focused on cucumber leaves, three wavelengths (4, 7.5, and 8.5 mm) of millimeter-wave irradiation were employed. The leaf biopotential under three different intensities of millimeter-wave exposure differentially altered [50]. In summary, irradiating plants with millimeter waves at different doses and durations provoked dynamic morphophysiological effects in plants, most of which were beneficial for the increase of productivity. To date, the responses of only a few plants under millimeter-wave irradiation have been investigated, leaving the majority to be investigated.

### 2.3. Proteomic Responses

To uncover regulatory mechanisms that led to changes of morphophysiological characteristics, proteomic techniques such as two-dimensional polyacrylamide gel electrophoresis (2-D PAGE) and gel-free/label-free analysis were used to illustrate protein changes in individuals treated with millimeter-wave irradiation. Some of published studies focused on animal cells such as melanoma cells [51] and macrophage cells [52], while none of the studies focused on plants irradiated with millimeter waves until a year ago, which analyzed the proteomic alterations on irradiated soybean that went through normal growth and flooding stress [24]. Recently, proteomic responses of another leguminous plant, chickpea, under millimeter-wave irradiation were analyzed [48]. These two studies, both of which used a gel-free/label-free proteomic approach, together built initial understanding of proteomic responses in plants under millimeter-wave irradiation.

One of the common responses in irradiated soybean and chickpea revealed by proteomic analysis was activated photosynthesis. In soybean, proteins related to photosynthesis, such as photosystem I P700 chlorophyll a apoprotein A2, cytochrome f, chlorophyll a-b binding protein, photosystem II CP47 reaction center protein, photosystem I reaction center subunit III, and chloroplast ATP synthase, increased in millimeter-wave-irradiated soybeans at 2- and 4-day old without flooding and 4-day old under flooding conditions [24]. In chickpea, ribulose-1,5-bisphosphate carboxylase/oxygenase (RuBisCO) activase and RuBisCO large subunit decreased with flooding stress, and they recovered with millimeter-wave irradiation; however, RuBisCO small subunit did not change under flooding, and it increased with irradiation [48]. RuBisCO composed of large and small subunits is the rate-limiting enzyme for photosynthetic carbon fixation [53], which is regulated by RuBisCO activase via ATP hydrolysis [54], and it has been proved to augment plant tolerance against drought, salinity, and heat [55,56]. These results indicated that millimeter-wave irradiation induced different changes on photosynthesis in plants, such as increase of proteins in photosystems in soybean and carbon fixation in chickpea; however, these results were not contradictory as the study on soybean focused on root-hypocotyl tissues and the study on chickpea focused on leaves. These studies co-explained positive regulation of millimeter-wave irradiation on plant photosynthesis, which might lead to morphophysiological changes.

Despite that millimeter-wave irradiation improved plant growth and flood tolerance in both of the two leguminous plants, several proteomic findings were distinct in different plants. Under a control condition where plants are not flooded, protein alterations during the plant development stage in irradiated soybeans were very different from un-irradiated ones, and chaperonin 10 significantly increased [24]; however, in chickpea, proteins were not largely changed between irradiated and un-irradiated plants at this stage [48]. The discordance of proteomic changes in different plants is reasonable, since different plants have a distinct tolerance to stress. Moreover, the cell tissues from soybeans and chickpeas used for proteomic analysis were different, which implies that tissue-specific or organ-specific responses of plants under millimeter-wave irradiation might exist and require further investigation. Furthermore, oppositely altered proteins between irradiated and un-irradiated soybeans under flooding were mainly related to sugar metabolism and the antioxidant system, while in the case of chickpeas, they were mainly related to fermentation and protein degradation [24,48]. The significantly altered proteins in these categories are displayed in a simplified schematic diagram (Figure 1), which were likely to determine flooding tolerance in irradiated plants. Among these categories, sugar metabolism, glycolysis, and fermentation formed an important pathway for energy production as flooding adaptation [57,58]. Dynamic proteomic responses in soybean and chickpea contributed to comprehensive understanding of regulatory mechanisms in plants under millimeter-wave irradiation. Currently, proteomic data of plants irradiated with millimeter waves are limited, which require investigation in the future.

## 3. Ultraviolet

### 3.1. Characteristics

UV lights refer to the part with shorter wavelength than human-visible lights in the solar spectrum. Although UV lights are invisible to the human eye, some insects can see them [59]. In the electromagnetic field, UV band positions are between the X-ray (200 nm) and visible light region (400 nm). UV lights are further subdivided into long-wave UV-A (320 to 400 nm), medium-wave UV-B (280 to 320 nm), and short-wave UV-C (200 to 280 nm) [60]. Different subtypes of UV have distinct patterns when coming across the stratospheric ozone layer. UV-A largely penetrated through the layer and reached the earth’s surface; UV-B was strongly absorbed in both the stratosphere and the lower atmosphere [61]; meanwhile, UV-C was completely screened out and could not reach the earth’s surface [62]. Because the atmosphere does little to shield UV-A, UV-A level at the earth’s surface seems stable; however, the levels of UV-B at the earth’s surface are influenced by factors such as time of year or latitude, depending on the ozone concentrations [63].

It was reported that emissions of the ozone-depleting substance damaged the global ozone layer, which increased exposure of the biosphere under sunlight, especially UV-B irradiation [64]. UV-B irradiation occupies a small fraction of total solar irradiation, yet it contains higher energy than UV-A and visible light according to the rule of Planck relation, and it elicits dynamic responses in various living organisms at varied exposure levels [65]. There have been numerous studies focused on biological effects and action mechanisms of UV-A and UV-B irradiations; meanwhile, since UV-C irradiation did not present as an environmental stress to the biosphere, less attention was paid to it [62]. Nevertheless, as the wavelength of UV-C is the shortest among the UV light region, it contains the highest energy, which excellently inactivates bacteria and viruses and has great potential in the manufacturing industry [66]. To date, knowledge about the responses of microorganisms under UV irradiation has long been elucidated [67], while UV irradiation-induced effects on plants are still under investigation.

### 3.2. Morphophysiological Effects

Generally, some plants such as coffee are naturally adapted to UV and continued to grow and produce under the increased environmental UV levels [68], owing to stocky phenotype and morphological traits of their leaf, such as increased stomatal density, epidermal thickness, and sunscreen accumulation in the leaf surface [69]. For most of the other plants, UV acted both as an environmental stress eliciting a stress-control response [70] and an informational development signal inducing photomorphogenic responses [71]. However, the morphophysiological effects of UV irradiation in plants varied on different subtypes of UV, duration of exposure, dose of exposure, as well as plant species, age, and other factors [72]. UV-A waves were mainly perceived by blue-light photoreceptors such as cryptochromes and induced photomorphogenic effects in plants [73]. It was revealed that UV-A had stimulatory effects on plant growth and production. In tomato, a supplement of UV-A increased leaf area, which facilitated light capture and stimulated plant biomass production [74]. In lettuce, a supplement of UV-A led to the increase of shoot dry weight [75]. In *Laurus nobilis*, a supplement of UV-A improved water-use efficiency via increased leaf relative water content, which led to the increase of leaf thickness and total biomass [76]. However, not all plants responded to UV-A irradiation with an increase of biomass, and inhibitory effects of UV-A irradiation in plants such as wheat [77], cucumber [78], and soybean [79] were also discovered, such as suppressed plant growth and declined biomass.

Comparing with UV-A irradiation, UV-B irradiation cannot reach as deep target sites in leaves as UV-A does due to the shorter wavelength; however, UV-B photons contain higher energy and lead to more intense impacts on plants. UV-B waves were perceived by UVR8 [80], which not only induced photomorphogenic effects but also stress-control responses under high-dose conditions. On the one hand, similar to UV-A, UV-B irradiation had a species-specific influence on photosynthesis and changed plant morphology [81]. On the other hand, excess absorption of UV-B by various biomolecules, especially DNA, resulted in their destructive damage, which triggered a stress-response mechanism that is similar to other abiotic oxidative stress [82]. These responses included over-production of reactive oxygen species [83], impairment of cell processes [84], as well as alteration of phytohormone metabolism and transport [85]. In addition, a recent study revealed that UV-B irradiation altered synthesis of UV-B reflecting pigments in leaves of *Zinnia*, which changed leaf color to acclimate UV-B irradiation [86].

Many studies focused on the effects of UV-A or UV-B irradiation, yet UV-C irradiation has received increasing attention. Like UV-B, UV-C irradiation can induce oxidative damage and evoke stress responses in plants [87]. The morphophysiological effects of UV-C irradiation in crops such as tomato [88], pepper [89], and strawberry [90] varied under different dosages among different species. Notwithstanding UV-C irradiation had similar effects as other UV irradiations on plants, its performance on crop post-harvest treatment was the best among UV irradiations to reduce microbial growth on plant surfaces and was beneficial for storage [91,92].

Despite distinct responses in plants under different subtypes of UV irradiation, the accumulation of secondary metabolites in plants was observed in all subtypes of UV irradiation. For plants themselves, accumulation of secondary metabolites with UV-absorbing properties such as flavonoids eased them from oxidative stress [93]. For humans, accumulation of secondary metabolites useful for health in plants improved their nutraceutical value, which has great prospects for industrial development [94]. The identified increasing secondary metabolites from different resource plants are summarized in Table 3. Of note, the accumulation of active compounds under UV irradiation has been discovered in several medicinal plants [95,96,97,98,99,100,101,102,103,104], which encouraged future investigation of the regulation mechanisms.

### 3.3. Proteomic Responses

To uncover underlying mechanisms regulating morphological and physiological alterations mentioned above, especially the accumulation of secondary metabolites under UV irradiation, numbers of comparative proteomic analyses in different plants were performed via a broad range of proteomic techniques. Under UV-A irradiation, protein responses in leaves of *Taxus chinensis* [100] and flower buds of *Lonicera japonica Thunb.* [119] have been investigated using 2-D PAGE. Under UV-C irradiation, protein profiles in leaves of *Cynara cardunculus* have been examined using 2-D PAGE and 2-D difference in-gel electrophoresis [120,121]. In the meantime, proteomic responses in plants under UV-B irradiation were the mostly studied, including the studies in *Arabidopsis thaliana* leaves using isobaric tags for relative and absolute quantitation (iTRAQ)-based analysis [122], *Catharanthus roseus* leaves using gel-free/label-free analysis [105,123], rice leaves using 2-D PAGE [124], Populus cathayana leaves using iTRAQ-based analysis [125], soybean seedlings [126]/sprouts using 2-D PAGE/iTRAQ-based analysis [127], Euphorbia kansui laticifers using iTRAQ-based analysis [128], and barley leaves using combinatory analysis of 2-D PAGE plus gel-free/label-free-based LC-MS/MS detection [106]. According to these proteomic analyses, the abundance of proteins related to photosynthesis, energy production/consumption, antioxidant reactions, and secondary metabolism under UV irradiations significantly altered (Figure 2). The signaling pathways for plant sensing UV were investigated as well [129], which were largely dominated by photoreceptor UVR8 (Figure 2). Furthermore, some of genes encoding proteins that related to these pathways such as cryptochrome in photosynthesis [130], phenylalanine ammonia-lyase in secondary metabolism [131], and NADPH-dependent thioredoxin reductase in antioxidant reaction [132], were manually modified in plants, after which the resistance/sensitivity under UV irradiation were changed.

Significant responses related to photosynthesis have been reported in many plants, where proteins involved in light reactions such as light harvesting complex II, photosystem II polypeptide, cytochrome b6/f, photosystem I reaction center subunit, and ferredoxin NADP^+^ oxidoreductase largely decreased [121,124,125]. Gene expression level of RuBisCO large subunit downregulated in *Lonicera japonica* under UV irradiation, which together suggested the inhibitory effects of photosynthesis by UV irradiation [119]. In energy production/consumption pathways, proteins underwent dynamic changes. For example, proteins and encoding genes belonging to mitochondrial electron transport chain complex I decreased, while those in complex II increased, which increased ATP content in *C. roseus* under UV-B irradiation [123]. Similarly, other proteins related to energy metabolism such as isocitrate dehydrogenase, V-type proton ATPase, and transitional endoplasmic reticulum ATPase increased in *L. japonica* under UV irradiation as well, where gene expression level of 6-phosphogluconate significantly upregulated [119]. However, in *Populus cathayana*, several ATP carrier- and exchange-related proteins decreased instead [125]. In the plant enzymatic antioxidant system, many proteins increased as protective responses under UV irradiation, including thioredoxin family protein, Fe superoxide dismutase, cytochrome B5 isoform, peroxidase, cysteine synthase, and glyoxalase [105,106,120,124]. Gene expression level of glyoxalase upregulated in rice under UV irradiation, which was consistent with the change of protein levels [124]. In plant secondary metabolism, many proteins as well as their gene expression levels altered in accordance with the increasing trend of secondary metabolites, which included phenylalanine ammonia lyase, chalcone synthetase, flavonoid synthetase, and terpenoid biosynthesis-related proteins [120,122,128]. In addition, activity of phenylalanine ammonia lyase was found to be increased in Lonicera japonica under UV irradiation, which was responsible for activation of the phenylpropanoid pathway [119].

Furthermore, it has been revealed that combination of UV-B irradiation with dark treatment enlarged the accumulative effects of secondary metabolites in leaves of medical plants such as *Catharanthus roseus* [99], *Lonicera japonica* [133], *Mahonia bealei* [134], and Clematis terniflora [135]. To clarify involving mechanisms, comparative proteomic analyses between the binary-stress treated groups and control groups of several plants using the gel-free/label-free technique were performed, which led to the discovery of much more altered proteins related to secondary metabolism under UV irradiation followed with dark treatment. In Catharanthus roseus, an abundance of 10-hydroxygeraniol oxidoreductase increased, which was related to the biosynthesis of indole alkaloid [99]. In Lonicera japonica, the abundance of 1-deoxy-D-xylulose 5-phosphate reductoisomerase and 5-enol-pyruvylshikimate-phosphate synthase increased, which promoted a supplement of precursors for caffeoylquinic acids and iridoids [119]. In Mahonia bealei, the abundance of S-adenosyl-L-methionine synthetase increased, which guaranteed high concentration of S-adenosyl-L-methionine for enhanced biosynthesis of benzylisoquinoline alkaloids in plant seedlings [136]. In Clematis terniflora, the abundance of proteins related to amino-acid metabolism such as S-adenosylmethionine synthetase, cysteine synthase, dihydrolipoyl dehydrogenase, and glutamate dehydrogenase increased, which led to antioxidant defense and accumulation of gamma-aminobutyric acid [111].

Thus far, integrative analyses combining transcriptomics and metabolomics with proteomics have been performed in some plants for comprehensive knowledge of regulatory mechanisms in plants under UV-B irradiation or UV-B irradiation with dark treatment. In Catharanthus roseus, gas chromatography-based and liquid-chromatography-based metabolomic studies were performed, which indicated that metabolites that related to the pentose phosphate pathway and amino acid metabolism altered under UV-B irradiation in accordance with changes of protein levels. These alterations regulated the flux of the methylerythritol phosphate pathway to the biosynthesis of monoterpene moieties and led to accumulation of various indole alkaloids [105,123]. In Mahonia bealei, liquid-chromatography-based untargeted/targeted metabolomic technique was used for investigation of metabolite changes under combined ultraviolet and darkness treatment, which illustrated changes of metabolites related to respiration, phenylalanine metabolism, and nitrogen metabolism [134]. Integration analysis of proteomics and metabolomics proposed that the citrate cycle played an important role in modulating the flux from 2-oxoglutarate to amino acid metabolism, which was linked to biosynthesis of down-stream secondary metabolites [134]. In Clematis terniflora, transcriptomic and metabolomic analyses under ultraviolet and darkness treatment were performed, where genes as well as proteins in pathways related to posttranslational modification, ubiquitin proteasome, and ribosomal protein largely changed [101,111]. In spite of these studies, changes of many proteins responsible for catalyzation of specialized secondary metabolites in plants remain undetected due to their low abundance. Further in-depth research, towards changes of proteins related to secondary metabolism in plants under UV-B irradiation, are needed.

## 4. Gamma Rays

### 4.1. Characteristics

Gamma rays belong to electromagnetic irradiations, which are released when the nuclear energy level transitions to deexcitation. They are produced by the hottest and the most energetic objects in the universe, such as neutron stars, pulsars, supernova explosions, and regions around black holes [137]. On earth, gamma rays are generated by nuclear explosions, lightning, and the less dramatic activity of radioactive decay. Gamma rays are the electromagnetic waves with the wavelength shorter than 0.01 angstrom [138]. Gamma rays were first discovered by French scientist P.V. Villard, belonging to the third kind of nuclear rays discovered after alpha and beta rays [139]. Unlike optical light and X-rays, gamma rays cannot be captured or reflected by mirrors [140]. Gamma-ray wavelengths are so short that they can pass through the space within the atoms of a detector. Gamma rays are the most energetic form of electromagnetic irradiation, having the energy level from around ten to several hundred kilo electron volts, and are more penetrating than other radiation such as alpha and beta rays [141].

Gamma rays belong to ionizing irradiation and interact with atoms or molecules to produce free radicals in cells. These radicals can damage or modify important components of plant cells and differentially affected the morphology, anatomy, biochemistry, and physiology of plants depending on irradiation level [141]. These effects include changes in plant cellular structure and metabolism, such as dilation of thylakoid membranes, alteration in photosynthesis, modulation of antioxidative systems, and accumulation of phenolic compounds [141,142]. Due to the above characteristics, gamma rays are usually applied for plant breeding.

### 4.2. Morphophysiological Effects

Seedlings exposed to a relatively low dose gamma rays (1–5 Gy) developed normally, while the growth of plants irradiated with a high dose gamma ray (50 Gy) was significantly inhibited [143]. At a subcellular level, chloroplasts, mitochondria, and endoplasmic reticulum were extremely sensitive to gamma irradiation [143]. Gamma irradiation mainly affected plant morphophysiology through regulating synthesis and scavenging of reactive oxygen species [143]. Morphophysiological effects of gamma irradiation on plants are summarized in Table 4.

In *Allium cepa*, when seedlings were treated with a gamma source at doses ranging from 0.1 to 10 Gy for 6 and 10 days, the growth of root and shoot was inhibited after 6 days exposure at all doses, including the low dose 0.1 Gy. However, at 10 days, growth inhibition of root and shoot was only observed after irradiation above 5 Gy [144]. When popular plantlets were concomitantly exposed to gamma rays at doses of 10, 20, 50, 100, 200, and 300 Gy, respectively, plant height, stem diameter, and biomass of seedlings were suppressed [148]. After receiving 200 and 300 Gy of gamma irradiation, all plantlets stopped growing, and most of them withered after 4–10 weeks of gamma irradiation [148].

Wheat seeds treated with a 20 Gy dose of gamma irradiation improved germination capacity compared to non-irradiated ones [149]. In maize, germination potential and root/shoot length of seedlings decreased with increasing irradiation doses. Plants derived from seeds irradiated with 500 Gy of gamma rays did not survive more than 10 days [150]. A low dose (50 Gy) of gamma rays stimulated germination and shoot growth initiation in *Lathyrus chrysanthus*; however, high doses of gamma irradiation inhibited seed germination and seedling growth [151]. When quinoa seeds were irradiated with a low dose of gamma irradiation, plant height and biomass significantly increased compared to the un-irradiated group [152]. In soybean, when seeds were irradiated with 200 Gy of gamma rays for 20 h, root growth was not suppressed even under flooding stress [26]. In short, morphophysiological effects of gamma irradiation on plants are dose dependent and low-dose irradiation has the potential to promote seed germination, plant growth, and stress tolerance.

### 4.3. Proteomic Responses

To understand underlying mechanisms inducing morphophysiological effects by gamma irradiation on plants, the proteomic approach was applied. In rice, a gel-based proteomic approach was used to investigate the effects of gamma irradiation on leaf metabolism. As a result, 59 gamma-irradiation-responsive proteins were identified and those related to cell metabolism were the most positively affected, while the photosynthesis process was negatively affected by low doses of gamma irradiation [153]. In *Chlamydomonas reinhardtii,* gel-based proteomic analysis indicated that gamma irradiation induced accumulation of proteins related to photosynthesis, carbon metabolism functions, and antioxidant functions, which enhanced lipid production [154].

Thus far, the flooding tolerance of gamma-irradiated soybean was systematically investigated using proteomic technique (Figure 3). A flooding-tolerant soybean mutant line was developed through gamma irradiation and alcohol dehydrogenase significantly increased in the mutant under flooding, indicating that the activation of the fermentation system was essential for gamma irradiation-induced flooding tolerance in soybean [26]. It was indicated that proteins related to protein synthesis and RNA regulation significantly changed in mutant soybean at initial flooding stress, and notably nascent polypeptide-associated complex (NAC), chaperonin 20, glycine-rich RNA-binding protein, as well as eukaryotic aspartyl protease increased at protein abundance and mRNA expression levels [23]. NAC contributes to assembly and transport of newly synthesized proteins and protects nascent polypeptides from proteolysis [155]. Chaperones play an important role in sustaining protein homeostasis such as protein folding, disaggregation, and degradation [156]. It has been reported that chaperone genes significantly contribute to *S. furcifera* tolerance to temperature and UV-A stress [157]. These findings indicate that refolding and assembly of newly synthesized proteins might be involved in gamma irradiation-mediated flooding tolerance in soybean.

Transcriptomic analysis indicated that RNA regulation and protein metabolism related genes were significantly changed in mutant soybean at initial flooding stress [158]. Among them, flooding tolerance negatively contributed to genes including *ATPase family AAA domain-containing protein 1*, *glucose-6-phosphate isomerase*, *matrix metalloproteinase*, and *cytochrome P450 77 A1* that were up-regulated in wild type soybean; however, they were returned to normal levels in the flooding-tolerant mutant line under flooding stress. Metabolomic analysis of the flooding-tolerant mutant and abscisic acid-treated soybeans suggested that accumulated fructose might play a role in initial flooding tolerance through regulation of hexokinase and phosphofructokinase [159]. An integration of proteomics and computational genetic modification effectiveness analysis indicated that energy-related proteins such as glyceraldehyde-3-phosphate dehydrogenase, aconitase 1, and 2-oxoglutarate dehydrogenase were higher in flooding-tolerant soybean [160]. Additionally, calreticulin specifically accumulated in flooding-tolerant soybean and regulation of cell death through the fermentation system/glycoprotein folding was an important factor for the acquisition of flooding tolerance [161]. In summary, gamma irradiation-induced flooding tolerance is a complex process, and cell wall metabolism, energy metabolism, protein synthesis, as well as transcriptional regulation-related proteins contributed to gamma irradiation-induced flooding tolerance in soybean.

## 5. The Effects on Abiotic Stress Tolerance of the Different Irradiation Sources

Several studies showed that plant seed pretreatments with microwave irradiation or visible light were an effective approach to ameliorate the hazards to seedling growth caused by abiotic stresses (Table 5). Weak microwave irradiation to wheat seeds within 20 s improved plant growth with longer root length and greater seedling weight compared to the control condition, and 10 s treatment further benefited plant growth under salt stress through stimulation of the antioxidant defense system [162]. Similar results were found when wheat seeds were treated with 10 s microwave irradiation followed by osmotic stress [163]. Furthermore, cadmium stress reduced antioxidant enzymes and antioxidative compounds in wheat seedlings compared with unstressed plants; however, seeds with microwave pretreatment for 5 or 10 s provided protection for wheat from cadmium-caused oxidative damage [164]. These reports uncovered that reactive oxygen species suppression involved in plant tolerance to abiotic stresses were provoked by microwave irradiation. Souza et al. [165] found although silver nanoparticles were internalized in plant cells during onion seed germination under dark and light conditions, the bigger aggregates and lower toxicity were presented in an 8 h light condition, implying that visible light reduced genotoxicity and cytotoxicity in plants. In summary, seed priming with suitable doses of microwave irradiation or visible light exposure could confer abiotic stress tolerance on plants; however, relative mechanisms underlying plant tolerance far beyond reactive oxygen species catabolism should be revealed, especially for biotic stress.

## 6. Conclusions and Future Perspectives

Over the past decades, knowledge of electromagnetic-wave irradiation on plants has been vastly improved. Herein, basic properties of electromagnetic spectral bands and plant responses to electromagnetic-wave irradiation have been simplified, including fluctuated metabolites, activated cellular metabolisms, and potential effects on crops (Figure 4). As discussed above, microwave and millimeter wave irradiation were mainly conducted as a seed priming approach, and ultraviolet irradiation was performed for seedling leaves. Meanwhile, more attention has been paid to leaf response to electromagnetic-wave irradiation compared with other organs, which was uncovered by omics analyses, showing that nuclei, mitochondria, and chloroplasts were targeted at the subcellular level and were involved in UV-provoked photosynthesis in young seedlings. Since the root is an underground organ, studies in which roots were directly irradiated with electromagnetic waves were limited. However, seed pretreatment with weak microwaves or millimeter wave irradiation made changes in root physiology with increased root length and weight, but ultraviolet irradiation posed opposite effects.

Current knowledge on plant responses to electromagnetic-wave irradiation were largely acquired from UV-B. UV-B induced signaling pathways and interactions between UV and lights in plants were drafted based on detection of *UVR8*, the specific photoreceptor for UV-B. Furthermore, interplay between UV-B and lights was proved with coordinated carbohydrates shifting dependent on diurnal variations, while documents addressing other electromagnetic waves coordinating with the circadian clock are rare. With continuous advancements, proteomics combined with metabolomics profiled responsive proteins and metabolites in plants exposed to electromagnetic-wave irradiation, which were largely related to carbohydrate provision, secondary metabolism, and redox homeostasis. Among these activated cellular metabolisms, much emphasis was placed on redox homeostasis; however, core components evoking signaling pathways need to be explored in more detail. Additionally, moderate doses of electromagnetic-wave irradiation enhanced plant resistance far beyond the decline of insects, and studies focusing on specific interactions within plants and microorganisms should be paid more attention with the aid of the microbiome. In future, proteomic approaches coupled with transcriptomics, metabolomics, metagenomics, and bioinformatics will largely facilitate studies of electromagnetic-wave irradiation on plants and the plant–microorganism context.

## Figures and Tables

**Figure 1 ijms-22-12239-f001:**
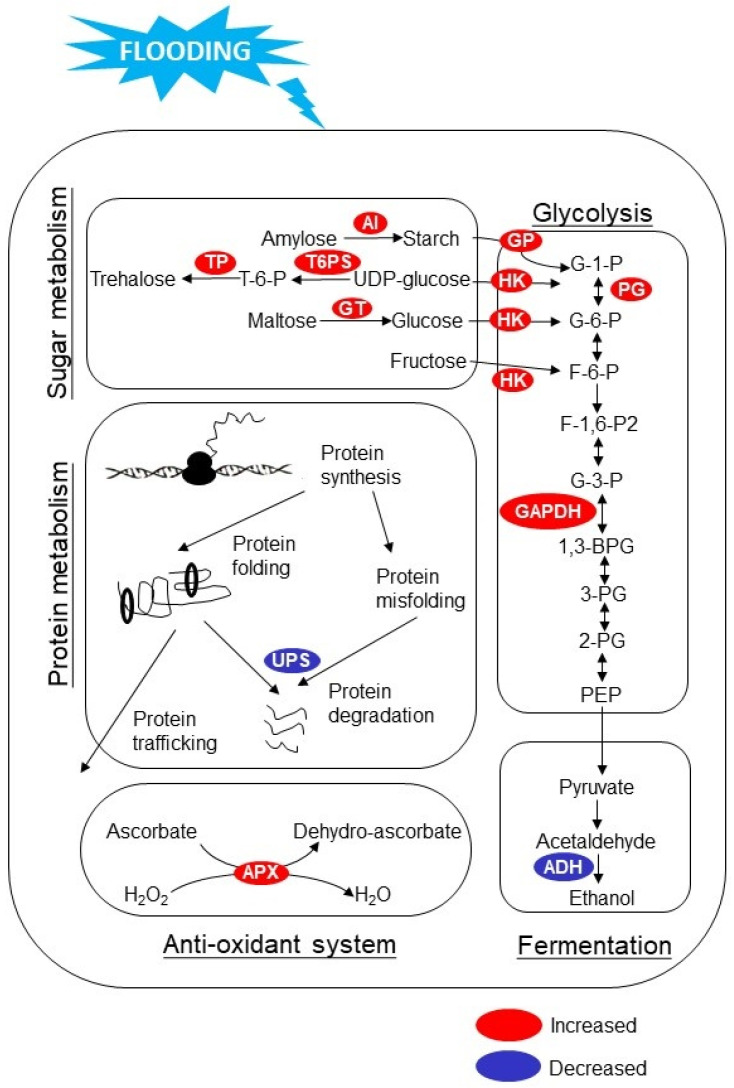
Altered proteins in millimeter-wave-irradiated plants compared to un-irradiated plants under flooding stress. The altered proteins according to previous studies were mapped onto a schematic diagram. Abbreviations are as follows: AI, amylose isomerase; TP, trehalose phosphatase; T6PS, trehalose 6-phosphate synthase; GT, glucanotransferase; GP, glycogen phosphorylase; HK, hexokinase; PG, phosphoglucomutase; GAPDH, glyceraldehyde-3-phosphate dehydrogenase; UPS, ubiquitin-proteasome system; APX, ascorbate peroxidase; ADH, alcohol dehydrogenase; T-6-P, trehalose-6-phosphate; G-1-P, glucose-1-phosphate; G-6-P, glucose-6-phosphate; F-6-P, fructose-6-phosphate; F-1,6-P2, fructose 1,6-bisphosphate; G-3-P, glyceraldehyde 3-phosphate; 1,3-BPG, glycerate 1,3-bisphosphate; 3-PG, 3-phospho-glycerate; 2-PG, 2-phospho-glycerate; PEP, phosphoenolpyruvate.

**Figure 2 ijms-22-12239-f002:**
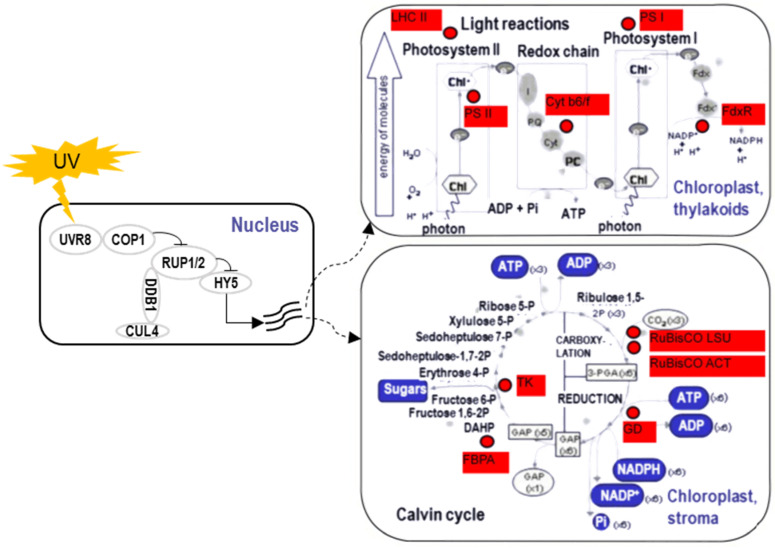
Signaling model in response to ultraviolet irradiation and altered proteins related to photosynthesis in plants. The altered proteins related to photosynthesis were mapped onto a pathway map from the MapMan software and marked with red color. Blue boxes indicate metabolites involved in photosynthesis. Curves in the left square indicate responsive transcripts. Abbreviations are as follows: LHC II, light harvesting complex II; PS II, photosystem II polypeptide; Cyt b6/f, cytochrome b6/f; PS I, photosystem I reaction center subunit; FdxR, ferredoxin NADP^+^ oxidoreductase; TK, transketolase; FBPA, fructose bisphosphate aldolase; GD, glyceraldehyde-3-P dehydrogenase; RuBisCO, ribulose-1,5-bisphosphate carboxylase/oxygenase; LSU, large subunit; ACT, activase; UVR8, UV RESISTANCE LOCUS 8; COP1, CONSTITUTIVELY PHOTOMORPHOGENIC 1; RUP, REPRESSOR OF UV-B PHOTOMORPHOGENESIS; DDB1, Damaged DNA binding protein 1; CUL4, Cullin 4; HY5, ELONGATED HYPOCOTYL 5.

**Figure 3 ijms-22-12239-f003:**
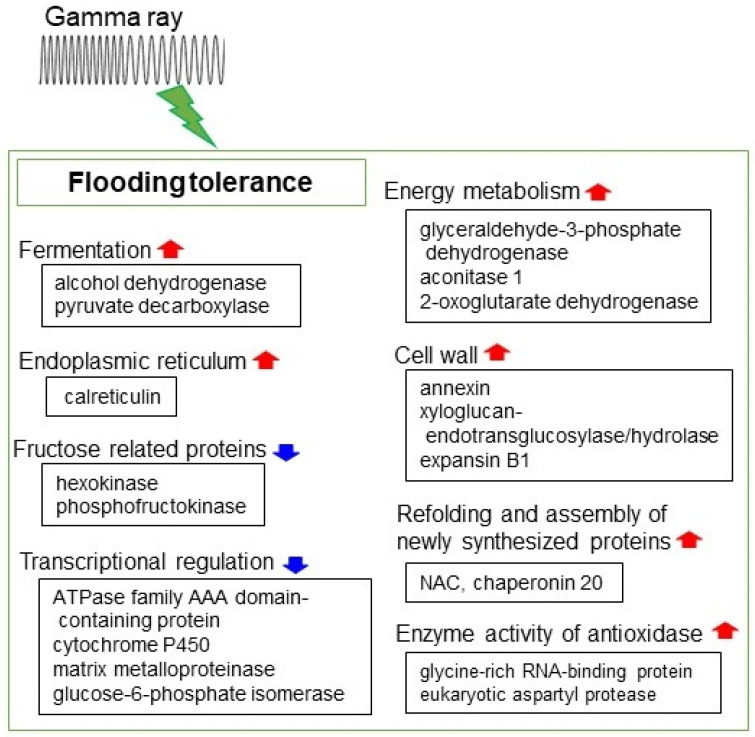
Summarizing overview of gamma irradiation-induced flooding tolerance in soybean. The upward red arrows mean activated metabolisms or increased proteins, while downward blue arrows mean suppressed metabolisms or decreased proteins.

**Figure 4 ijms-22-12239-f004:**
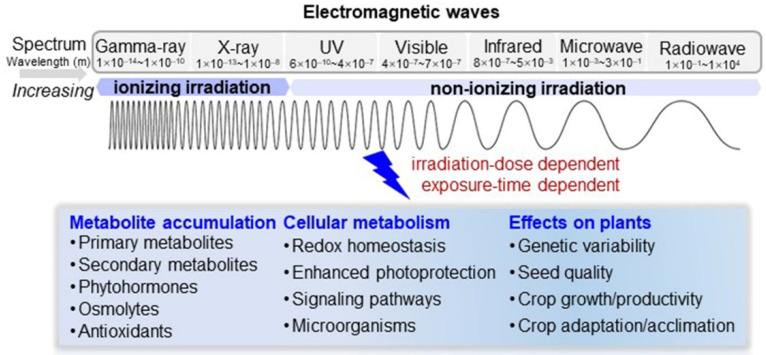
Summarizing overview of electromagnetic-wave irradiation on plants. Physical properties of spectral bands of electromagnetic waves are indicated. Electromagnetic-wave irradiation induced metabolite shifts and cellular metabolisms that were associated with plant growth and environment adaptation are summarized based on published studies on plants irradiated with electromagnetic waves.

**Table 1 ijms-22-12239-t001:** Characteristics of electromagnetic waves.

Electromagnetic Spectrum	Wavelength (m) ^a^	Frequency (Hz) ^a^	Source to Emit Spectra ^b^	Radioactive Categories
audio/radio waves	1 × 10^−1^–1 × 10^4^	3 × 10^4^–3 × 10^9^	obtained with a ferro or piezoelectric transducer	non-ionizing irradiation
microwaves	1 × 10^−3^–3 × 10^−1^	1 × 10^9^–3 × 10^11^	emitted by a magnetron or a klystron	
infrared	8 × 10^−7^–5 × 10^−3^	6 × 10^10^–4 × 10^14^	emitted by an incandescent object	
visible light	4 × 10^−7^–7 × 10^−7^	4 × 10^14^–7 × 10^14^	emitted by an electric light bulb	
ultraviolet	6 × 10^−10^–4 × 10^−7^	7 × 10^14^–5 × 10^17^	radiated with deuterium or mercury vapor lamps	
X-ray	1 × 10^−13^–1 × 10^−8^	1 × 10^16^–3 × 10^21^	emitted when electrons collide on a metal plate	ionizing irradiation
gamma ray	1 × 10^−14^–1 × 10^−10^	3 × 10^18^–3 × 10^22^	emitted by radioactive elements	

^a^ Wavelength and frequency of electromagnetic waves are referred from Einstein [3]; ^b^ sources emitting special electromagnetic waves are referred from Lewandowski [1].

**Table 2 ijms-22-12239-t002:** Morphophysiological effects of millimeter-wave irradiation on crops.

Plant Species	Morphophysiological Effects	Ref ^b^
Soybean	increased hypocotyl length/weight and main root length	[24]
Wheat	increased fresh weight, shoot height, length of main ear, number of grains in an ear, grain weight in an ear, lipid-peroxidation rate, catalase activity, malondialdehyde content, and flood tolerance; improved germination rate and germination potential; altered water absorption during germination; shortened phenophase	[41,42,43,44,45]
Brown rice	stimulated germination; increased polyphenol content and DPPH ^a^ radical scavenging activity; decreased gamma-aminobutyric acid content	[46]
Chickpea	increased leaf length/weight, root length/weight, and flood tolerance; decreased cell death under flooding	[48]

^a^ DPPH, 1,1-diphenyl-2-picrylhydrazyl; ^b^ Ref, References.

**Table 3 ijms-22-12239-t003:** Morphophysiological effects and accumulation of secondary metabolites in plants under ultraviolet irradiation.

Plant Species	UV-Subtype	Morphophysiological Effects	Accumulated Secondary Metabolites	Ref ^b^
Mung bean	UV-B	increased activities of phenyl alanine ammonia-lyase, L-galactono-1, 4-lactone dehydrogenase, and chalcone isomerase	vitamin C; total phenolics; total flavonoids	[94]
*Ginkgo biloba*	UV-B	unknown	total flavonoids; quercetin; kaempferol	[95]
*Astragalus membranaceus* Bge.	n.s. ^a^	decreased chlorophyll content, stomatal conductance, and net photosynthesis rate; increased activities of superoxide dismutase, catalase, and ascorbate peroxidase	calycosin-7-O-beta-D-glucoside; daidzein; calycosin	[96,104]
*Lonicera japonica* Thunb.	UV-A, UV-B	increased antioxidant activity	chlorogenic acid; 3,4-di-O-caffeoylquinic acid; 3,5-di-O-caffeoylquinic acid; 4,5-di-O-caffeoylquinic acid; secologanic acid; secoxyloganin; secologanin; (E)-aldosecologanin	[97]
*Artemisia annua*	UV-B	decreased contents of chlorophyll/carotenoid, photosynthetic rate, stomatal conductance, and transpiration rate; increased activities of RuBisCO	essential oils	[98]
*Catharanthus roseus*	UV-B	increased ATP content in leaves	strictosidine; vindoline; catharanthine; ajmalicine	[99,105]
*Taxus chinensis*	UV-A	damaged structures of chloroplasts and mitochondria	paclitaxel; 10-deacetylbaccatin III; baccatin III	[100]
*Achyranthes bidentata* Blume	UV-B	decreased plant height, root length, fresh weight of aerial parts/roots, and contents of photosynthetic pigments; increased activities of superoxide dismutase and peroxidase	oleanolic acid; ecdysterone	[102]
*Salvia miltiorrhiza* Bunge	UV-B	unknown	salvianolic acid B; rosmarinic acid; danshensu	[103]
Barley	UV-B	decreased elongation rate of primary roots and root osmotic pressure; increased modulus of elasticity of roots and cell wall rigidity	saponarin	[106,107]
Birch	UV-B	unaffected leaf morphology	quercitrin; myricetin-3-galactoside; chlorogenic acid	[108]
Broccoli	UV-B	increased resistance against insect feeding	kaempferol; quercetin; glucosinolates	[109]
*Centella asiatica*	UV-B	decreased content of chlorophyll; increased absorbance of adaxial epidermises at 375 nm, and necrotic spots on the epidermises	kaempferol-3-O-beta-d-glucuronopyranoside; quercetin-3-O-beta-d-glucuronopyranoside	[110]
*Clematis terniflora*	UV-B	decreased leaf area and biomass; increased occurrences of burned patches and crispation in leaves	luteolin 7-O-beta-D-glucosiduronic acid; rutin; kaempferol 3-O-rutinose	[111]
Grape berry	UV-C	increased relative mass of skins; unaffected berry weight and berry caliber	trans-resveratrol; piceid; viniferin	[112]
*Polygonum cuspidatum*	UV-C	unknown	resveratrol	[113]
*Psychotria brachyceras*	UV-B	unknown	brachycerine	[114]
Radish	UV-A	decreased plant height;increased release of hydrogen	anthocyanin	[115]
Rice	n.s. ^a^	decreased leaf photosynthetic rate, pollen germination, spikelet fertility, and yield; increased spikelet abortion	N-trans-cinnamoyltryptamine; N-(p-coumaroyl) serotonin; N-cinnamoyltyramine	[116,117]
Willow	UV-B	increased shoot biomass	luteolin-7-glucoside; monomethyl-monocoumaryl-luteolin-7-glucoside; myricetin derivative; apigenin-7-glucuronide; p-hydroxycinnamic acid derivative	[118]

^a^ n.s., Not specified; ^b^ Ref, References.

**Table 4 ijms-22-12239-t004:** Morphophysiological effects of gamma irradiation on plants.

Plant Species	Treatment of Gamma Irradiation	Effects	Ref ^b^
Soybean	Seeds were irradiated with 200 Gy of gamma rays for 20 h.	Root growth was not suppressed even after being exposed to flooding stress for 4 days.	[26]
Onion	Seedlings were irradiated at doses ranging from 0.1 to 10 Gy of a ^137^Cs gamma source for 6 and 10 days. ^a^	The growth of root and shoot was inhibited after 6 days exposure at all doses, including the low dose (0.1 Gy). At a later point in time (day 10), root and shoot inhibition was observed after irradiation at high doses (above 5 Gy).	[144]
*Cymbidium hybrid*	Cymbidium hybrid RB001 protocorm-like bodies were irradiated in a time course and dose-dependent manner (1 h, 16.1 Gy; 4 h, 23.6 Gy; 8 h, 37.9 Gy; 16 h, 37.9 Gy; and 24 h, 40.0 Gy) of gamma rays.	Based on survival rate of the plant, the estimated optimal doses were duration-dependent at irradiation durations shorter than 8 h.	[145]
Cowpea	Seeds were irradiated by ^60^Co source with dose of 11 kGy and the actual dose delivered was an average of 11.2 kGy at a dose rate of 1.7 kGy h^−1^.	Irradiation led to decrease in wall thickness, increase of cell size, and intercellular spaces in cotyledon.	[146]
Common vetch	Seeds were irradiated with 100 Gy of gamma irradiation.	Irradiation pretreatment (100 Gy), alone or in combination with salt stress and drought stress, led to significant increases in dry matter accumulation, catalase/superoxide dismutase/ascorbate peroxidase activities, and proline contents. However, gamma-irradiation pretreatment alone increased chlorophyll contents while decreasing malondialdehyde contents.	[147]
Poplar	Plantlets were concomitantly irradiated at doses of 10, 20, 50, 100, 200, and 300 Gy, respectively (dose rates ranged from 0.5 to 15 Gy h^−1^), for 20 h in ^60^Co.	Acute irradiation with a dose of 100 Gy greatly reduced height, stem diameter, and biomass of poplar plantlets. After receiving doses of 200 and 300 Gy, all plantlets stopped growing, and most of them withered after 4–10 weeks of irradiation.	[148]
Wheat	Seeds were irradiated at doses of 0, 10, 20, and 30 Gy.	The 20 Gy dose improved seed germination capacity compared with non-irradiated ones.	[149]
Maize	Seeds were irradiated at doses ranging from 0.1 to 1 kGy of gamma rays.	Germination potential and physiological parameters of maize seedlings decreased by increasing irradiation dose. Plants derived from seeds exposed at higher doses (0.5 kGy) did not survive more than 10 days.	[150]
*Lathyrus chrysanthus*	Seeds were irradiated with different doses (0, 50, 100, 150, 200, and 250 Gy) of ^60^Co at 0.8 kGy h^−1^.	Low dose irradiation stimulated germination and shoot growth initiation; however, high level irradiation inhibited seed germination and seedling growth.	[151]
Quinoa	Seeds were irradiated at 50, 100, and 200 Gy emitted by ^60^Co.	Plant height and biomass increased in quinoa treated with a low dose (50 Gy) compared to the control.	[152]

^a^ Gy, Grays; ^b^ Ref, References.

**Table 5 ijms-22-12239-t005:** The effects on abiotic stress tolerance of the different irradiation sources.

Plant/ Organs	Treatment	Stress	Finding	Ref ^b^
Wheat/ Seeds	microwave irradiation at 2.45 Ghz for 10 s	Salt	Low energy microwave irradiation pretreatment of seeds for 10 s protected seedlings from salt stress by enhanced enzyme activities of nitric oxide synthase, catalase, peroxidase, superoxidase dismutase, and glutathione reductase.	[162]
Wheat/ Seeds	microwave irradiation at 2.45 Ghz for 10 s	Osmotic	Microwave irradiation of seeds for 10 s conferred plant tolerance to osmotic stress by enhancing nitric oxide signaling and antioxidant defense system.	[163]
Wheat/ Seeds	microwave irradiation at 2.45 Ghz for 5, 10, and 15 s	Cd	Seeds pretreated with microwave irradiation for 5 or 10 s ameliorated plant growth under Cd stress by decreasing lipid peroxidation and hydrogen peroxide accumulation.	[164]
Onion/ Seeds	fluorescent lamp exposure with 32 w for 8 h	AgNPs ^a^	Light exposure reduced genotoxicity and cytotoxicity of AgNPs by reducing uptake of NPs by plant cells.	[165]

^a^ NPs, Nanoparticles; ^b^ Ref, References.

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
