# Peer review of "Morphophysiological and Proteomic Responses on Plants of Irradiation with Electromagnetic Waves"

_ijms, 2021, doi:10.3390/ijms222212239_

Round 1
Reviewer 1 Report
This review addresses and summarizes the findings available in the literature in relation to the morphophysiological and proteomic changes in plants as a result of the irradiation with different types of electromagnetic waves. The review offers a valuable and comprehensive overview of the impact of waves in three different electromagnetic spectrums: millimeter waves, UV and gamma ray. Nevertheless, the manuscript would benefit from a major revision of the English language used.
A few comments/questions,
1) I would recommend including a clear explanation of why the review focuses on millimeter waves, UV and gamma ray, and not any of the other electromagnetic waves described in Table 1.
2) In my opinion, the paper lacks sufficient information about the proteomic techniques used in the different proteomic studies analyzed. Proteomics is a quite broad term, that can include gel-based or gel-free approaches, quantitative label-free or with chemical labels, and a broad range of analysis/statistical approaches that can determine that a protein appears or not as differentially expressed. The review would benefit from the inclusion of such information.
For example, in lines 183-185, a study in soybeans show differentially expressed proteins upon millimeter-wave irradiation while another study in chickpea did not show significant changes. As a reader interested in proteomics, I wonder if any of the technical aspects might help to explain this difference.
3) In line 315, reference 117 appears repeated and with the wrong format.
4) I would recommend making the red/blue arrows bigger, as it is difficult to see in the image that these are arrows and if they are going up or down.
Author Response
Reviewer 1
This review addresses and summarizes the findings available in the literature in relation to the morphophysiological and proteomic changes in plants as a result of the irradiation with different types of electromagnetic waves. The review offers a valuable and comprehensive overview of the impact of waves in three different electromagnetic spectrums: millimeter waves, UV and gamma ray. Nevertheless, the manuscript would benefit from a major revision of the English language used.
Answer: Thank you very much for your comments and suggestion. As suggested, all parts have been corrected with red color. Furthermore, current revised version has been checked by native speaker of English.
A few comments/questions,
1) I would recommend including a clear explanation of why the review focuses on millimeter waves, UV and gamma ray, and not any of the other electromagnetic waves described in Table 1.
Answer: Thank you very much for your recommendation. The explanation has been added to the text as suggested in “Introduction” section with red color. In brief, effects of microwave, visible light, and X-ray on crop physiology have been reported, which focused on changes in morphology and physiology; however, the corresponding metabolisms and signaling response were uncertain. Besides, we focus to review plant proteomic response to electromagnetic wave irradiation, which have already been carried out for millimeter wave, UV, and gamma ray.
2) In my opinion, the paper lacks sufficient information about the proteomic techniques used in the different proteomic studies analyzed. Proteomics is a quite broad term, that can include gel-based or gel-free approaches, quantitative label-free or with chemical labels, and a broad range of analysis/statistical approaches that can determine that a protein appears or not as differentially expressed. The review would benefit from the inclusion of such information.
Answer: Thank you for the suggestion. The information about the proteomic techniques used in cited studies has been reviewed and added in proteomic parts.
For example, in lines 183-185, a study in soybeans show differentially expressed proteins upon millimeter-wave irradiation while another study in chickpea did not show significant changes. As a reader interested in proteomics, I wonder if any of the technical aspects might help to explain this difference.
Answer: The explanation has been added to the text as suggested. In brief, the discordance in proteomic changes between soybean and chickpea might be explained by difference in plant material. Firstly, the growth and development of different plants are distinct, which could lead to different stress tolerance. More importantly, the tissue parts that used for proteomic analysis of soybeans and chickpeas were different. In the study of soybeans, proteins from root-hypocotyl tissue were extracted and analyzed, while in the study of chickpeas, proteins from leaves were extracted and analyzed. Tissue-specific or organ-specific responses of plants under millimeter-wave irradiation might exist and require further investigation.
3) In line 315, reference 117 appears repeated and with the wrong format.
Answer: I am sorry we made a mistake. It has been corrected.
4) I would recommend making the red/blue arrows bigger, as it is difficult to see in the image that these are arrows and if they are going up or down.
Answer: Thank you very much for your point out. They have been corrected in Figure 3.
Reviewer 2 Report
In this review, the authors address the effects of electromagnetic waves on plants. Although the topic is very interesting, the authors should pay attention to some relevant points to improve the quality of the manuscript and its readability.
- Figure 2. Please indicate what blue/red colour stands for.
- Is there any plant species naturally adapted to irradiation with electromagnetic waves? What are their morphological traits?
- Are there genetically modified plants more resistant/susceptible to irradiation? What genes are essential for each source?
- How does the plant sense the irradiation? Are there receptors? If so, summarize in a graph for each source how is the signalling pathway for each of them.
- Do plants have an increased number of mutations? How is the radiation affecting the genetic responses?
- Please also show a table for UV light with morphophysiological traits in different plant species.
- It is not clear why the authors focus on flooding stress tolerance and not other stresses.
- Please summarize the effects on biotic and abiotic stress tolerance of the different irradiation sources in a table. In that sense, address one section of the review to tackle the impact on plant stress tolerance.
- It is unclear why the authors decided to focus on protein levels for this review and ignore the other "omics". The authors base the concept of "proteomic" responses on protein levels without addressing the potential posttranslational effects or the enzymatic activities. In general, the authors have put little effort into interpreting the experimental data available and delivering the (digested) information to the reader. Instead of writing a list of protein names for every irradiance source, please tell what routes are activated and combine this information with metabolite and gene expression levels. Increased protein levels alone mean nothing.
- What is the cell tissue targeted by irradiance? What cells are most sensitive? What plant stage? What are the effects at the root level?
Author Response
Reviewer 2
In this review, the authors address the effects of electromagnetic waves on plants. Although the topic is very interesting, the authors should pay attention to some relevant points to improve the quality of the manuscript and its readability.
1) Figure 2. Please indicate what blue/red colour stands for.
Answer: Thank you very much for your correction. The information for blue and red colors has been indicated in figure legends of Figure 2.
2) Is there any plant species naturally adapted to irradiation with electromagnetic waves? What are their morphological traits?
Answer: As millimeter wave and gamma ray are virtually absent from natural environment, the knowledge about plants naturally adapted to such electromagnetic waves is largely unknown, while there are certain plant species found with acclimation strategies to avoid stress symptoms under ultraviolet. For examples, despite the negative impacts of increased ultraviolet radiation intensity on plants, coffee plants continued to grow and produce under the increased environmental UV levels (Bernado et al., 2021). The stocky phenotype and the morphological traits of their leaves are important for adaptation, such as increased stomatal density, epidermal thickness, and sunscreens accumulation in leaf surface, which are well-documented in the chapter ‘Plant Survival Under Natural UV Radiation on Earth: UV Adaptive/UV-Adapted Traits’ in book Natural UV Radiation in Enhancing Survival Value and Quality of Plants (Sen Mandi, 2016). This answer has been added to the manuscript in red color in UV section.
Reference
Bernado, W. de P. et al. Biomass and leaf acclimations to ultraviolet solar radiation in juvenile plants of Coffea arabica and C. Canephora. Plants 2021, 10, 640.
Sen Mandi, S. Natural UV Radiation in Enhancing Survival Value and Quality of Plants; Springer India: New Delhi, 2016; ISBN 978-81-322-2765-6.
3) Are there genetically modified plants more resistant/susceptible to irradiation? What genes are essential for each source?
Answer: Thank you for the question. After substantial literature reviewing, genetically modified plants with altered resistance/sensitivity under UV irradiation were found, while none for millimeter-wave and gamma ray irradiations. For UV irradiation, some genes encoding proteins in responsive pathways including photosynthesis (Khudyakova et al., 2019), antioxidant reaction (Bashandy et al., 2009), and secondary metabolism (Huang et al., 2010) are confirmed to be essential for the resistance/sensitivity by gene modification experiment. The information has been added to the manuscript in red color in UV section.
Reference
Khudyakova, A.Y. et al. Impact of UV-B radiation on the photosystem II activity, pro-/antioxidant balance and expression of light-activated genes in Arabidopsis thaliana hy4 mutants grown under light of different spectral composition. J. Photochem. Photobiol. B Biol. 2019, 194, 14–20.
Bashandy, T. et al. Accumulation of flavonoids in an ntra ntrb mutant leads to tolerance to UV-C. Mol. Plant 2009, 2, 249–258.
Huang, J. et al. Functional analysis of the Arabidopsis PAL gene family in plant growth, development, and response to environmental stress. Plant Physiol. 2010, 153, 1526–1538.
4) How does the plant sense the irradiation? Are there receptors? If so, summarize in a graph for each source how is the signalling pathway for each of them.
Answer: Among the electromagnetic irradiations, only the signaling pathway for UV has been studied, which was well documented in Jin and Zhu’s review (2019). In addition, working model for plant sensing UV irradiation has been summarized in a graph and integrated into Figure 2.
Reference
Jin, H., Zhu, Z. Dis-RUP for COP1 Role-Switch under UV-B Light. Trends Plant Sci. 2019, 24, 569–571.
5) Do plants have an increased number of mutations? How is the radiation affecting the genetic responses?
Answer: Ultraviolet and ionizing radiations caused DNA damage by induction of reactive oxygen species, which may cause lethal mutation and threaten plant survival (Gill et al., 2015). In the meanwhile, DNA-repair mechanisms occur in radiated plants, which get rid of certain damages (Waterworth et al., 2011). However, as mentioned in manuscript, the effects on these radiations largely depend on dosage and species. Moreover, the repair ability of different species varies from each other. Besides, study of genetic responses requires constant evaluation on the offspring, which is not an easy job. As a result, there might lack conclusive answer for plants do have an increased number of mutations under radiations or not.
Reference
Gill, S.S. et al. DNA damage and repair in plants under ultraviolet and ionizing radiations. Sci. World J. 2015, 2015, 5–7.
Waterworth, W.M. et al. Repairing breaks in the plant genome: The importance of keeping it together. New Phytol. 2011, 192, 805–822.
6) Please also show a table for UV light with morphophysiological traits in different plant species.
Answer: Thank you for the suggestion. The morphophysiological traits of UV light have been added to Table 3.
7) It is not clear why the authors focus on flooding stress tolerance and not other stresses.
Answer: Thanks for your comment. Effects of different irradiation sources on plant stress tolerance have been added to the manuscript in section 5.
8) Please summarize the effects on biotic and abiotic stress tolerance of the different irradiation sources in a table. In that sense, address one section of the review to tackle the impact on plant stress tolerance.
Answer: Thank you very much for the comments. New section of “5. The Effects on Abiotic Stress Tolerance of The Different Irradiation Sources” has been added and Table 5 has been prepared in revised manuscript. Because we could not find publications describing positive effects on plant resistance to biotic stress with other different irradiation sources, we did not summary the effects on biotic stress tolerance.
9) It is unclear why the authors decided to focus on protein levels for this review and ignore the other "omics". The authors base the concept of "proteomic" responses on protein levels without addressing the potential posttranslational effects or the enzymatic activities. In general, the authors have put little effort into interpreting the experimental data available and delivering the (digested) information to the reader. Instead of writing a list of protein names for every irradiance source, please tell what routes are activated and combine this information with metabolite and gene expression levels. Increased protein levels alone mean nothing.
Answer: Thank you for your comments. The information of protein posttranslational modification, enzymatic activity, metabolite and gene expression have been reviewed and integrated into the manuscript with the proteomic data for better readability. However, the research progress is different among different irradiance source. Effects of ultraviolet on Catharanthus roseus, Mahonia bealei, and Clematis terniflora have been explained by integrated omics, from which activated route has been interpreted by gene expression level, protein modification, and metabolite accumulation. In the meanwhile, the available data for millimeter-wave irradiation is rather limited, which largely focused on morphophysiological effects and lack such information to determine exact activated routes. Furthermore, proteomics is the large-scale analysis of difference at the protein level, which is highly associated with gene expression level and modulation of protein activity. In addition, proteomic data that cited in the present review were evaluated, where data confirmed by immunoblot analysis were interpreted and delivered to the reader. Therefore, proteomic data for each irradiance source helped to build elementary understanding of response mechanisms that potential leads to morphophysiological effects, and enabled comparisons of plants under different electromagnetic irradiations. Based on this notion, this review focused on proteomics, but also with indication of other omics for comprehensive conclusion.
10) What is the cell tissue targeted by irradiance? What cells are most sensitive? What plant stage? What are the effects at the root level?
Answer: Thanks for these questions. Description of sensitive cell tissue and subcellular organelles to electromagnetic wave irradiation has been added. In addition, root response to different irradiances has been summarized. These corrections have been added in section “6. Conclusion and Future Prospective” with red color.
Round 2
Reviewer 2 Report
The authors have addressed all my previous concerns and modified the manuscript accordingly. The manuscript is now ready for publication. As a minor correction, please adjust the duplication "b) b)" present at the end of Table 4.
Author Response
The authors have addressed all my previous concerns and modified the manuscript accordingly. The manuscript is now ready for publication. As a minor correction, please adjust the duplication "b) b)" present at the end of Table 4.
Answer: Thank you very much for your kind review. I am sorry we made a mistake. It has been corrected.